# Preparation and Characterization of Rutin–Loaded Zein–Carboxymethyl Starch Nanoparticles

**DOI:** 10.3390/foods11182827

**Published:** 2022-09-13

**Authors:** Cuicui Li, Long Chen, David Julian McClements, Xinwen Peng, Chao Qiu, Jie Long, Hangyan Ji, Jianwei Zhao, Xing Zhou, Zhengyu Jin

**Affiliations:** 1School of Food Science and Technology, Jiangnan University, 1800 Lihu Road, Wuxi 214122, China; 2Collaborative Innovation Center of Food Safety and Quality Control in Jiangsu Province, Jiangnan University, 1800 Lihu Road, Wuxi 214122, China; 3Department of Food Science, University of Massachusetts, Amherst, MA 01003, USA; 4State Key Laboratory of Pulp and Paper Engineering, South China University of Technology, Guangzhou 510640, China

**Keywords:** zein, carboxymethyl starch, rutin, nanoparticle, interaction, formation mechanism

## Abstract

In this work, rutin (RT)–loaded zein–carboxymethyl starch (CMS) nanoparticles were successfully prepared by the antisolvent precipitation method. The effect of CMS on composite nanoparticles at different concentrations was studied. When the ratio of zein–RT–CMS was 10:1:30, the encapsulation efficiency (EE) was the highest, reaching 73.5%. At this ratio, the size of the composite nanoparticles was 196.47 nm, and the PDI was 0.13, showing excellent dispersibility. The results of fluorescence spectroscopy, FTIR, XRD, and CD showed that electrostatic interaction, hydrogen bonding, and hydrophobic interaction were the main driving forces for the formation of nanoparticles. It can be seen from the FE–SEM images that the zein–RT–CMS nanoparticles were spherical. With the increase in the CMS concentration, the particles gradually embedded in the cross–linked network of CMS (10:1:50). After RT was loaded on zein–CMS nanoparticles, the thermal stability and pH stability of RT were improved. The results showed that zein–CMS was an excellent encapsulation material for bioactive substances.

## 1. Introduction

Dietary polyphenols are widely distributed in vegetables, fruits, and medicinal plants, and have attracted extensive attention due to their biological activities [1,2]. Flavonoids are the main category of dietary polyphenols [3]. Rutin (RT), also known as vitamin P, is a kind of flavonoid that widely exists in citrus fruits (2.7–8106.7 µg/g), apples (350–4780 μg/g), and tea (303–479 μg/g) [4,5,6,7]. RT has good antioxidant, free radical scavenging, anti–inflammatory, anti–cancer, antibacterial, cardiovascular, and neuroprotective properties [8]. However, RT is unstable due to its polyhydroxy structure, which is easily affected by environmental factors such as heat (>75 °C) and pH (strong acid and alkaline conditions); in addition, both the benzene ring and hydroxyl group exist in one molecular structure, resulting in poor solubility of RT [9]. Therefore, the application of RT in the food industry is limited. In view of the above shortcomings, encapsulation technology has gradually received attention in the food industry. At present, various liposomes, emulsions, micelles, and particles are used to encapsulate bioactive components [10,11,12,13]. The preparation of liposomes, emulsions, and other systems requires the addition of a large number of organic solvents and surfactants, which may be incompatible with the food industry. Nanoparticles have been proven to be excellent materials for encapsulating polyphenols. Nanoparticle encapsulation systems are not only simple to operate, but also can achieve high encapsulation efficiency and high solubility with or without the usage of organic solvents, which has attracted widespread attention [14,15].

The choice of encapsulation materials is very important for the stability of bioactive components. In recent years, natural biopolymers (proteins, polysaccharides, etc.) have been widely used as encapsulation carriers for bioactive components because of their safety, non–toxicity, biodegradability, and biocompatibility [16,17]. Food–grade proteins, such as soy protein isolate, casein, gelatin, and zein, have been used as packaging materials [18,19,20]. Zein is the main storage protein of corn, which has excellent self–loading characteristics [21]. Therefore, nanoparticles loaded with hydrophobic bioactive substances can be constructed by using self–loading properties [22]. The results showed that hydrophobic polyphenol–loaded protein nanoparticles could significantly improve the stability and solubility of polyphenols [23,24]. However, nanoparticles with a single protein have strong surface hydrophobicity, which makes the particles vulnerable to environmental impact and easy to aggregate [25]. Natural polysaccharides are considered to be effective stabilizers for zein carriers, such as gum, xanthan gum, and alginate. Muhammad et al. [26] used a zein–pectin carrier to load quercetin, which improved the thermal stability and photostability of quercetin. However, the high viscosity and low charge density of some polysaccharides limit their application [27]. Therefore, more polysaccharides that can be used to stabilize particles needs to be explored. Starch is the most common carbohydrate, and is a polysaccharide formed by the aggregation of glucose molecules. Its cost is low and it is easy to modify at the molecular level. Carboxymethyl starch (CMS) is a modified anionic starch ether. Compared with the native starch and other polysaccharides, it has high hydrophilicity, electronegativity, weak retrogradation, high transparency, and freeze–thaw stability [28]. Some studies showed that CMS also had low sensitivity to bacteria or heat, which was due to the steric hindrance of the formation of macromolecular chains in the structure of CMS [29]. In addition, CMS–based drug delivery systems have been studied. For example, Saboktakin et al. [30] studied chitosan–CMS nanoparticle carriers to transport 5–aminosalicylic acid to the colon; Zhang et al. [31] alternately deposited CMS and cationic quaternary ammonium salt starch to prepare nano–capsules, which could further deliver proteins to the digestive tract. However, there are few reports on the preparation of nanoparticles by zein and CMS, which is a direction worthy of further exploration.

In this study, RT–loaded zein–CMS nanoparticles were prepared by the antisolvent precipitation method. The particle size, polymer dispersion index (PDI) and ζ–potential were studied. In addition, the encapsulation efficiency (EE) and loading capacity (LC) of zein–CMS nanoparticles for RT were studied. Finally, the structure and physicochemical properties of zein–CMS nanoparticles and zein–CMS nanoparticles loaded with RT were characterized. The results of this study may help to develop a protein–polysaccharide delivery system for bioactive ingredients so as to expand the application of bioactive ingredients in the food industry.

## 2. Materials and Methods

### 2.1. Materials

Zein (purity ≥90%, average molecular weight: 25,000~45,000) was acquired from Macklin (Shanghai, China). Rutin (purity ≥95%), and dimethyl sulfoxide (purity ≥98%) was purchased from Sinopharm Chemical Reagent Co. LTD. (Shanghai, China). Carboxymethyl starch (purity ≥95%, degree of substitution 0.3–0.6) was acquired from Aladdin (Shanghai, China). All other reagents were analytically pure.

### 2.2. Preparation of RT–Loaded Zein–CMS Nanoparticles

Composite nanoparticles were prepared by the antisolvent precipitation method based on the work of Liu et al. [32] with some modifications. First, CMS was dissolved in deionized water and stirred for 2 h as a stock solution. Zein (0.2 g) was added to 20 mL of 75% ethanol solution with agitation at 600 rpm for 1 h, then RT (0.02 g) was added and stirred in the dark for another hour; the mixed solution of zein and RT was dropped into the aqueous solution of CMS at a volume ratio of 1:4 and agitated at 600 rpm for 1 h in the dark. The ethanol in the sample was removed by rotating the evaporator (Temperature of 40 °C, vacuum of 0.1 MPa) and the volume was supplemented with deionized water. Then, the sample with the supplemented volume was centrifuged at 1000× *g* for 10 min at 4 °C to remove unembedded RT and macromers. Finally, the freshly prepared samples were subjected to freeze–drying, and then the dried samples were further analyzed. The samples were prepared at room temperature conditions (temperature of 25 °C, humidity level of 40–50%).

### 2.3. Determination of Particle Size, Polydispersity Index (PDI), ζ–Potential, and Turbidity

The particle size, polydispersity index (PDI), and ζ-potential of the nanoparticles were measured by a Zeta–sizer nano ZS (Malvern company, Malvern City, UK). The sample used for the measurement was diluted 10 times with deionized water (pH = 4) to ensure that the particle size and the ζ–potential of the samples was accurate. The freshly–prepared sample was diluted 2 times, then its turbidity was measured at 600 nm using a UV–vis spectrophotometer.

### 2.4. Characterization of the Complex

#### 2.4.1. Fluorescence Spectrum Analysis

The fluorescence of the raw materials (zein, RT, and CMS) and samples (zein, zein–RT and zein–RT–CMS nanoparticles) were analyzed through the fluorescence spectrophotometer (F–7000, Hitachi, Tokyo, Japan) using the method described in the work of Dai et al. [33] with some modifications. The measured samples were diluted to a proper concentration using deionized water (pH = 4). The excitation wavelength was set to 280 nm, the acquisition wavelength range was 290–450 nm, and the scan speed was 240 nm/min. Excitation and emission bandwidths were set to 5 nm. All data were collected at room temperature conditions (temperature of 25 °C, humidity level of 40–50%).

#### 2.4.2. Fourier–Transform Infrared (FTIR) Spectroscopy

The FTIR spectra of the raw materials (zein, CMS, and RT) and samples (zein–RT and zein–RT–CMS nanoparticles) were analyzed by the FTIR spectrometer (IS10, Bruker, Billerica, MA, USA). The samples were mixed with potassium bromide (KBr) at a ratio of 1:100 and grounded until there were no visible particles, then pressed into transparent circular flakes (circular flakes was 13 mm and the thickness was 0.1–0.5 mm), which were determined by FTIR. The scanning range of FTIR was 400 cm^−1^–4000 cm^−1^.

#### 2.4.3. X–ray Diffraction (XRD)

The crystal structures of the raw materials (zein, RT, and CMS) and samples (zein, zein–RT, and zein–RT–CMS nanoparticles) were determined through XRD (D2 PHASER, Germany Brock AXS Co., Ltd, Karlsruhe, Germany). The 2θ angle was from 5° to 45°.

#### 2.4.4. Circular Dichroism (CD) Spectroscopy

The structures of zein were determined by CD (Chirascan V100, Applied Optical Physics, Dublin, UK). The samples (zein and zein–RT–CMS nanoparticles) were diluted to a concentration of 0.2 mg/mL with deionized water (pH = 4). Under constant nitrogen flushing, the far–UV region was 180–260 nm, the pathlength in the far–UV region was 0.1 cm, the recording speed was 40 nm/min, and the bandwidth was 1 nm.

#### 2.4.5. Field Emission Scanning Electron Microscopy (FE–SEM) Analysis

The microstructures of the freeze–dried samples (zein, zein–RT, and zein–RT–CMS) were observed using FE–SEM (SU8100, Hitachi High–Tech Co., Tokyo, Ltd. of Japan) at an accelerating voltage of 5 kV. Before observation, the sample (2–3 mg) was uniformly smeared on the stage and then sprayed with gold [34].

### 2.5. Encapsulation Efficiency (EE) and Loading Capacity (LC) of Curcumin

The evaluation of the encapsulation efficiency and loading capacity of bioactive substances was one of the evaluation indicators for measuring the performance of the carriers. Freshly–prepared samples (2 mL) were centrifuged (4 °C, 12,000× *g*) for 30 min to acquire the liquid supernatant. The supernatant (600 μL) was evenly mixed with dimethyl sulfoxide (DMSO). Then, its absorbance at 364 nm was measured by UV–vis spectrophotometer. The amount of RT in the samples was determined using a calibration curve established with a standard solution (0–32 ug/mL free RT in DMSO), and the resulting standard curve is y = 0.0534x, R² = 0.9994. The EE and LC of RT were calculated according to the following formulas:$$EE\left(\%\right)=\frac{total\;RT-free\;RT}{total\;RT}\times 100$$
$$LC\left(\%\right)=\frac{total\;RT-free\;RT}{total\;amount\;of\;complex}\times 100$$

### 2.6. Stability of Complex

#### 2.6.1. Thermogravimetric Analysis (TGA) 

The thermal stability of the raw material and nanoparticles was determined by TGA (TGA2, Mettler Toledo Instruments Co., Ltd., Zurich, Switzerland). The samples (<3 mg) were added to the crucible and analyzed at a nitrogen flow rate of 10 °C/min and the temperature range of 30–600 °C.

#### 2.6.2. pH Stability

The pH stability of the samples was determined according to the method of Jiang et al. [35]. The pH value of the samples (10 mL) was adjusted to 2–8 with 1 M HCl or 1 M NaOH. The particle sizes, PDI, and ζ–potential of the samples was determined using the Zeta–sizer nano ZS (Malvern Company, Malvern City, UK).

### 2.7. Statistical Analysis

All experiments were repeated for at least three groups. The data are expressed as the mean ± standard deviation. SPSS–20 software was used for significance analysis, and the difference was significant when *p* < 0.05.

## 3. Results and Discussion

### 3.1. Particle Size, PDI, ζ–Potential, and Turbidity

The change in particle size, PDI, and ζ–potential affects the stability of the nanoparticles. Figure 1A,B showed the particle size, PDI, and ζ–potential of the nanoparticles (zein, zein–RT, and zein–RT–CMS nanoparticles). The particle size of pure zein nanoparticles was 104.10 nm [36]. The particle size of the zein nanoparticles loaded with RT was 104.77 nm. Compared with pure zein nanoparticles and zein–RT nanoparticles, the particle size of the zein–CMS nanoparticles loaded with RT increased significantly, as shown in Figure 1A. Moreover, with the increase in the CMS concentration, the particle size gradually increased from 159.13 to 289 nm, which was similar to the study by Wang et al. [37]. The possible reasons for this phenomenon were as follows: on the one hand, the increase in the CMS concentration resulted in the formation of a thick coating on the surface of the zein [38]; on the other hand, a large amount of CMS may interfere with the formation of nanoparticles [22]. PDI is an important index to characterize the dispersion of nanoparticles [39]. It can be seen from Figure 1B that the PDI of zein–RT–CMS nanoparticles was better than that of zein–RT nanoparticles. The PDI of zein–RT–CMS nanoparticles was less than 0.3, while the PDI of zein–RT nanoparticles was 0.45, which indicated that the addition of CMS can significantly improve the dispersion of nanoparticles. In addition, the PDI of zein–RT–CMS (10:1:30) nanoparticles was the minimum. A possible reason is that CMS might be evenly wrapped on the surface of zein at the ratio of 10:1:30, thus showing good dispersion [40].

Electrostatic interaction plays an important role in protein–polysaccharide complexes. In Figure 1B, the ζ–potential of zein nanoparticles was +31.63 mV when the pH was 4. The potential of CMS was −20.8 mV. With the increase in the CMS concentration, the ζ–potential value became negative, indicating that zein and CMS may be bound through electrostatic interaction. Hu et al. [16] also reported a similar phenomenon, wherein zein and pectin form complexes through electrostatic binding. Moreover, the negative ζ–potential of the complex also decreased with the increase in the CMS concentration, indicating that the ζ–potential of the complex was mainly controlled by CMS [41]. The absolute value of the ζ–potential of the ternary complex was between 20–35 mV, indicating that the particles had good stability [42].

Figure 1C shows the turbidity of the samples. Generally speaking, the turbidity depends on the particle size, concentration, and refractive index of particles [43]. It can be seen from Figure 1C that the turbidity of pure zein nanoparticles was the lowest. There were two possible reasons: first, it is related to the particle size. The smaller the particle size, the lower the turbidity. From the previous discussion, it can be seen that the particle size of pure zein nanoparticles was the smallest; second, it is related to refractive index. The lower the refractive index, the lower the turbidity. At the same temperature, the lower the concentration, the lower the refractive index. The concentration of pure zein nanoparticles was the lowest compared with the dispersion with foreign substances. When the level of CMS was between 10:1:10–10:1:40, the turbidity of the samples was proportional to the level of CMS, which may be due to the formation of large–sized nanoparticles after the addition of CMS. Under higher levels of CMS (10:1:50), the turbidity of the samples decreases. A possible reason was that the excess CMS cannot be combined with zein, and the unbound CMS was finally dissolved in the dispersion in a soluble state [44]. Chen et al. [45] also found a similar phenomenon. When the zein–hyaluronic acid ratio was 100:25 and 100:30, the turbidity of zein–hyaluronic acid complex decreased. 

### 3.2. Characterization of Nanoparticles

#### 3.2.1. Fluorescence Spectrum Analysis

The fluorescence signal emitted by proteins can be used to evaluate the interaction between additives and proteins [46]. A large number of tryptophan and tyrosine residues are presented in zein, which are the two main fluorescent groups at the 280 nm wavelength [47]. As shown in Figure 2, under the excitation wavelength of 280 nm, the fluorescence emission peak of zein was at 308 nm, which was similar to the study by Liu et al. [22]. According to Figure 2A, the addition of CMS increased the fluorescence intensity of zein, which may be due to the combination of CMS with the hydrophilic groups of zein, thereby exposing the tryptophan residues in zein [45]. However, after encapsulating polyphenols, the fluorescence intensity of zein–RT nanoparticles and zein–RT–CMS nanoparticles decreased, which may be caused by molecular rearrangement, energy transfer, and collision quenching [48]. Figure 2B shows the relationship between the fluorescence intensity and the level of CMS. At a low level of CMS (10:1:10–10:1:20), the fluorescence intensity of zein–RT–CMS nanoparticles was stronger than the zein nanoparticles alone. It may be that the addition of CMS resulted in the exposure of tryptophan residues in zein, and CMS cannot completely adhere to the surface of zein particles, thereby enhancing the fluorescence. When the level of CMS (10:1:30–10:1:50) increased, the fluorescence intensity of the complexes decreased. The possible reason was that excess CMS was wrapped on zein–RT nanoparticles, which resulted in the aggregation of particles and shielding of fluorophores [45]. However, when the zein–RT–CMS ratio was 10:1:30, the fluorescence intensity was the lowest. A possible reason was that more RT was encapsulated in zein–CMS nanoparticles because RT can reduce the polarity around tryptophan. Therefore, the fluorescence intensity of the complex was significantly reduced. This was similar to the study by Chen et al. [49].

#### 3.2.2. Fourier–Transform Infrared (FTIR) Spectroscopy

FTIR is generally used to study the interaction between molecules in nanoparticles and can effectively identify characteristic functional groups (Figure 3). The characteristic functional group is associated with the characteristic peaks in the spectrum. The interaction between molecules can be inferred from the shape, position, and intensity of the peaks in the spectrum [50]. As shown in Figure 3A, for zein, the characteristic peaks were 3310.45 cm^−1^ (O − H stretching vibration of hydroxyl), 1654.34 cm^−1^ (amide Ι band, mainly C = O stretching), and 1546.12 cm^−1^ (amide ΙΙ band, related to C − N stretching and N − H bending mode) [51]. For RT, the characteristic peaks were 3417.58 cm^−1^ (O − H stretching vibration), 1656.92 cm^−1^, 1601.49 cm^−1^ (mainly C = O stretching), 1504.74 cm^−1^ (C = C, aromatic), 1361.55 cm^−1^ (C − O (phenolic group)), and 1203.78 cm^−1^ (C – O − C). Other studies have reported similar results [52]. For CMS, the characteristic peaks were 3432.42 cm^−1^ (O − H stretching vibration), 1599.19 cm^−1^ (distribution of νCO in COO− group), 1420.31 cm^−1^ (− CH_2_ scissoring), and 995.52 cm^−1^ (aliphatic ether group C – O − C in CMS) [53]. Compared with zein, the O − H vibration stretching peak of zein–RT nanoparticles moved from 3310.45 cm^−1^ to 3303.31 cm^−1^, indicating the hydrogen bond interaction between zein and RT. In addition, the characteristic peaks of RT between 1000 cm^−1^ and 1500 cm^−1^ almost disappeared, indicating that RT was successfully encapsulated [54]. The O – H vibration stretching peak of zein–RT–CMS nanoparticles moved from 3310.45 cm^−1^ to 3411.17 cm^−1^, indicating that there was a strong hydrogen bond between zein and CMS [55]. Moreover, the characteristic peak of CMS appeared in the characteristic peak of the zein–RT–CMS nanoparticles, and the amide band also changed significantly, indicating that there was an electrostatic interaction between zein and CMS [25]. Furthermore, since zein and RT are highly hydrophobic molecules, hydrophobic interactions also occur during the formation of ternary complexes. The study of Meng et al. [54] also confirmed this assumption. 

From Figure 3B, with the increase of CMS concentration (10:1:10–10:1:50), the characteristic peaks of O – H bands were 3407.2 cm^−1^, 3412.98 cm^−1^, 3411.17 cm^−1^, 3413.12 cm^−1^, and 3422.27 cm^−1^, respectively. This similar phenomenon was also found in the study of Chen et al [45]. In addition, the characteristic peaks of zein–RT–CMS nanoparticles at the amide band, especially at the amide ΙΙ band, gradually disappeared, indicating that more CMS were wrapped on the surface of zein–RT nanoparticles. It is noteworthy that the zein–RT–CMS nanoparticles have characteristic peaks at 1024.89 cm^−1^, 1023.41 cm^−1^, 1022.27 cm^−1^, 1021.79 cm^−1^, and 1021.85 cm^−1^, and the peak intensity of these characteristic peaks became more and more significant. It was speculated as follows: on the one hand, CMS maybe interacted with RT molecules, which the aromatic ring of RT reacting with the aliphatic ether group of CMS; on the other hand, there may be non–covalent binding between zein and CMS, which may be the interaction between the C – O – C bond in CMS and zein. The above two reasons synergistically promote the generation of characteristic peaks (1024.89 cm^−1^, 1023.41 cm^−1^, 1022.27 cm^−1^, 1021.79 cm^−1^, and 1021.85 cm^−1^) and make the peak intensity more and more significant.

#### 3.2.3. X–ray Diffraction (XRD)

XRD is often used to analyze the crystallinity of samples. Figure 4 shows the XRD of the raw materials and samples. According to Figure 4A, zein had wide diffraction peaks at 9.3° and 19.5°, indicating that zein was amorphous. This was similar to the research by Sun et al [56]. The XRD of CMS showed that its main diffraction peaks were 15°, 17°, 18.3°, 23°, and 32.1°, which proved that it was a polycrystalline with a certain crystal form [57]. RT showed many diffraction peaks at 5–45°, which proved that it was crystalline [58]. From the XRD of zein–RT nanoparticles, it can be seen that the diffraction peak of RT disappeared, which indicated that RT was successfully encapsulated and became amorphous [22]. This was consistent with the results of FTIR. However, the zein–CMS nanoparticles loaded with RT showed the characteristic peak of CMS at 32.1° and the other characteristic peaks of CMS disappeared, indicating the interaction between CMS and zein–RT nanoparticles [40], which agreed with the FTIR analysis. As far as we know, the peak intensity of XRD is positively correlated with the height [59]. From Figure 4B, with the increase of CMS concentration, the characteristic peak height of zein–RT–CMS nanoparticles at 32.1° tended to increase first and then decrease, which also indicated that the characteristic peak strength of nanoparticles at 32.1° showed a trend of first increasing and then decreasing. When the ratio of zein–RT–CMS was 10:1:30, it was found that the peak intensity of zein–RT–CMS nanoparticles was the largest, indicating that the interaction between zein–RT–CMS was strong. This was consistent with the results of the fluorescence spectrum analysis.

#### 3.2.4. Circular Dichroism (CD) Spectroscopy

CD is often used to characterize the secondary structure changes of proteins. Figure 5 shows the changes in the secondary structure of zein. Zein had a positive peak at 195 nm, two negative peaks at 204 nm and 223 nm, and a zero crossing at 202 nm, which were the characteristic secondary structures of zein [60]. A CDNN program was used to calculate the content of α–helices, β–sheets, β–turns, and unordered coils of zein (Table 1). The α–helices, β–sheets, β–turns, and unordered coils of zein were 20.7%, 28.37%, 17.93%, and 36.67%, respectively. When RT and CMS were added, the contents of α–helical and unordered coils showed an upward trend, but the contents of the β–sheets and β–turns were opposite. With the increase in the CMS concentration (10:1:10–10:1:50), α–helices decreased from 23.70% to 21.47% and unordered coils decreased from 40.77% to 40.33%, while β–sheets increased from 25.33% to 28.17% and β–turns increased from 16.70% to 17.20%. This trend was similar to that of Sun et al [61]. Some studies have found that the increase in β–sheets may promote protein aggregation [62]. Therefore, when high–level CMS was present, the aggregation phenomenon of the complex may also be caused by the increase in β–sheets [63]. In addition, the increase in α–helices could make the secondary structure of the protein more compact [44]. The structural expansion and recombination of zein were caused by the addition of RT and CMS. This may be related to their non–covalent interaction, which can also be confirmed by fluorescence spectroscopy and FTIR. In addition, there may be cross–linking between higher levels of CMS, which may affect the conformation of the composite particle [64].

#### 3.2.5. Field Emission Scanning Electron Microscopy (FE–SEM) Analysis

Figure 6 shows the FE–SEM image of the zein, zein–RT, and zein–RT–CMS nanoparticles. Zein nanoparticles were typically spherical, which was consistent with previous reports (Figure 6a) [65]. It can be seen from Figure 6b–g that the morphology of zein–RT–CMS nanoparticles also showed a spherical form. Through the analysis of particle size, fluorescence spectrum, FTIR, and XRD, it can be seen that RT was successfully encapsulated in the particles, and CMS combined with zein nanoparticles by non–covalent binding force to form a spherical shape with smooth surface [61]. As can be seen from Figure 6g, at higher levels of CMS, self–cross–linking of CMS was generated. It was interesting that with the increase in the CMS concentration, zein–RT–CMS nanoparticles were embedded in the cross–linking network of CMS. A similarly interesting phenomenon was also found in the study by Chen et al. [49]. In this study, the increase in hyaluronic acid caused the nanoparticles to gradually transform into reticular gel. 

### 3.3. Encapsulation Efficiency (EE) and Loading Capacity (LC) of RT

The EE and LC of RT are shown in Figure 7. It can be seen from Figure 7 that the EE of zein–RT nanoparticles was 51.52%, which was lower than that of zein–RT–CMS nanoparticles. This may be due to the following reasons: Firstly, the non–covalent interaction between zein, RT, and CMS improved the EE of RT, which can be seen from the analysis of fluorescence spectroscopy, FTIR, and XRD [54]; secondly, zein and CMS have a synergistic effect on RT encapsulation [66]. In addition, when the ratio of zein–RT–CMS was 10:1:30, EE was 73.5%, which was the highest. A possible reason is that at this ratio, the non–covalent interactions between zein, RT, and CMS were the strongest, leading to the improvement of the EE. Similar results were also obtained in the study by Liu et al. [67]. In Liu’s study, when the carboxymethyl cellulose was increased, the EE of the composite also showed a trend of first increasing and then decreasing. In addition, we can see that the LC of RT gradually decreases, contrary to the trend of EE. This may be because the higher the content of CMS, the higher the total mass of nanoparticles, which was similar to the results from Li et al. [68].

### 3.4. Stability of Composite Nanoparticles

#### 3.4.1. TGA

TG is mainly used to evaluate the weight loss of samples with temperature [69]. Figure 8 shows the thermal stability of the raw materials and samples at 30–600 °C. As can be seen from Figure 8A, the thermal decomposition of RT can be divided into two main stages (30–300 °C). The first step was between 30–150 °C, and its mass loss was about 8.53%, which may be caused by the loss of water and the breakage of the single–bond molecular chain of RT [70]. The mass loss of the second step was about 22.58% at 200–300 °C, which may be caused by the loss of some chemisorption water and the breaking of most C–O bonds in the structure [71]. There were two main degradation stages for zein–RT nanoparticles and zein–RT–CMS nanoparticles. One was the mass loss caused by water loss in the range of 30–150 °C, and the other was caused by the pyrolysis and fracture of the molecular structure in the stage of 200–600 °C [72]. The mass loss of zein–RT nanoparticles was about 78.60% between 200–600 °C, while the zein–RT–CMS nanocomposites was 62.28%. The decrease in mass loss indicated that the thermal stability of the composite particles was improved after the addition of CMS [73]. The pyrolysis process was further analyzed by the mass change rate of the thermal analysis process (Figure 8B). RT has a maximum thermal degradation rate at 255 °C, and the maximum degradation peak shifts from 255 °C to 293 °C with the addition of CMS. The maximum degradation peak of zein–RT–CMS shifted backward compared with the pure CMS. It showed that the interaction between the ternary complex increases the maximum thermal degradation temperature. Although the maximum thermal degradation peak of zein–RT was 345 °C, its mass loss was more than that of the ternary complex, thereby indicating that the ternary complex was more helpful for improving the stability of RT. It was further explained that the addition of CMS improved the thermal stability of RT [74]. This result showed that zein–CMS, as an encapsulation carrier of polyphenols, could expand the application of polyphenols in the hot–processed food industry (such as frying, baking, cooking, etc.).

#### 3.4.2. pH Stability

The core–shell structure of protein–polysaccharides is easily affected by pH. The particle size, PDI, and ζ–potential can intuitively show the stability and dispersion of composite particles, which is also the most common characterization means [75]. Figure 9 shows the difference in particle size, PDI, and ζ–potential between zein–RT nanoparticles and zein–RT–CMS nanoparticles at pH 2–8. It can be seen from Figure 9 that zein–RT nanoparticles gathered and settled at pH 5–8, especially near the PI of zein (pH 5–6). This was mainly due to the increase in pH, which made the electrostatic repulsion between zein almost disappear, thereby leading to aggregation and precipitation. It was fully explained by the change of potential in Figure 9C. In addition, the process was irreversible [76]. On the contrary, the nanoparticles with CMS showed good stability and dispersion in a wide range of pH 3–8 (Figure 9). When the pH value was 2–3, zein–RT–CMS resulted in aggregation and sedimentation, especially when the pH value was 2, its particle size was greater than 1000 nm, and PDI was 0.92. This may be due to its ζ–potential value tending to 0, thereby leading to a sharp decrease in the electrostatic repulsion force between nanoparticles, which led to a decrease in the stability and dispersion of particles (Figure 9C). In addition, under strong acid conditions, the properties of CMS may be changed, which may also lead to particle aggregation [77]. With the increase in pH, it was found that the dispersion and stability of zein–RT–CMS nanoparticles remained at a good level (particle size < 550 nm, PDI < 0.45, ζ–potential absolute value > 25 mV).

## 4. Formation Mechanism of Composite Nanoparticles

The possible formation mechanism of zein–RT–CMS nanoparticles is proposed in Figure 10. Under the condition of magnetic stirring, zein and RT dissolved in 75% aqueous ethanol solution were dropwise added into the solution of CMS. Subsequently, the ethanol in the dispersion was removed by rotary evaporation. The formation of composite nanoparticles was mainly driven by hydrogen bonds, electrostatic interaction, and hydrophobic interaction, which was confirmed by the analysis of fluorescence spectra, FTIR, and XRD. Through the characterization of the properties and structures of the composite nanoparticles, it can be inferred that the level of CMS has a certain impact on the formation of nanoparticles. With the increase of CMS, the particle size and the absolute ζ–potential of the nanoparticles was increased. At a low level of CMS (10:1:10–10:1:20), CMS was not enough to be fully loaded on the surface of nanoparticles, resulting in weak non–covalent binding and poor stability. When the ratio of zein–RT–CMS was 10:1:30, the particle surface was completely covered by CMS, showing better EE and stability. Due to the high adhesiveness of CMS itself, as the level of CMS gradually increases (10:1:40), the particle size and PDI of composite nanoparticles increased. At the high level of CMS (10:1:50), cross–linking began to appear between CMS, which reduced the dispersion of nanoparticles and made them easier to aggregate. In addition, the particles changed from a typical spherical structure to a reticular structure filled with particles, which was proven by the FE–SEM. Moreover, zein–CMS nanoparticles have a higher encapsulation efficiency and better stability than pure zein nanoparticles.

## 5. Conclusions

In this study, RT–loaded zein–CMS nanoparticles were successfully prepared by the antisolvent precipitation method. The effects of CMS on the composite nanoparticles were evaluated. Through the particle size, ζ–potential, and PDI characterization of the composite nanoparticles, it was found that CMS improved the stability and dispersion of the composite nanoparticles, especially when the ratio of zein–RT–CMS was 10:1:30. Moreover, the EE of zein–CMS as an encapsulation carrier for RT was better than that of zein alone. In addition, the structural characteristics of the composite nanoparticles showed that there were hydrogen bonds, electrostatic interactions, and hydrophobic interactions among zein, RT, and CMS. The morphological characteristics of the composite nanoparticles were studied, and it was found that the composite nanoparticles presented a typical spherical shape. Furthermore, the formation mechanism of zein–RT–CMS nanoparticles was proposed according to the structural and morphological characterization. In the presence of higher–level CMS, nanoparticles gradually filled the cross–linked network of CMS. Finally, through the study of the stability of the composite nanoparticles, it was found that the thermal stability and pH stability were improved by the addition of CMS. In summary, these results suggested that zein–CMS as an encapsulation carrier not only made up for the shortcomings of easy aggregation of corn zein as an encapsulation carrier alone, but also improved the EE and the stability of the encapsulation material. This study provides a new idea for the loading of other active substances. In addition, this carrier (zein–CMS) in hot–processing food is also worth exploring in the future. 

## Figures and Tables

**Figure 1 foods-11-02827-f001:**
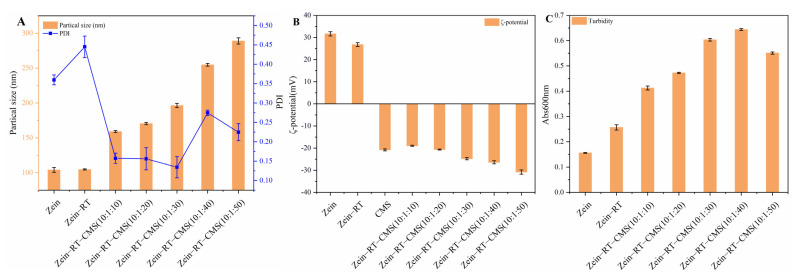
(**A**) Particle size and PDI of zein, zein–RT and zein–RT–CMS nanoparticles (at different levels of CMS); (**B**) ζ–potential of zein, zein–RT and zein–RT–CMS nanoparticles (at different levels of CMS); (**C**) turbidity of zein and zein–RT–CMS nanoparticles (at different levels of CMS).

**Figure 2 foods-11-02827-f002:**
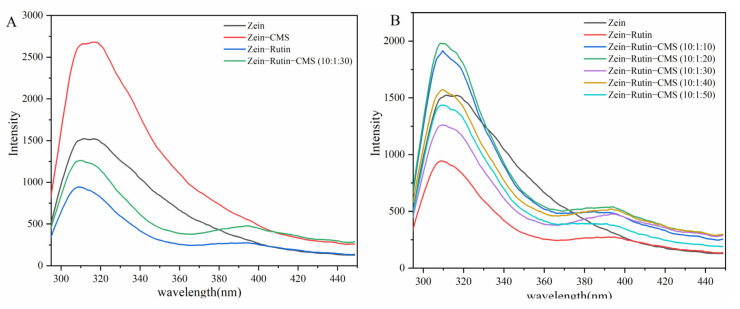
Fluorescence spectroscopy (**A**) Comparison of fluorescence intensity among zein, zein–CMS, zein–RT, and zein–RT–CMS nanoparticles; (**B**) Fluorescence intensity of zein–RT–CMS nanoparticles at different levels of CMS.

**Figure 3 foods-11-02827-f003:**
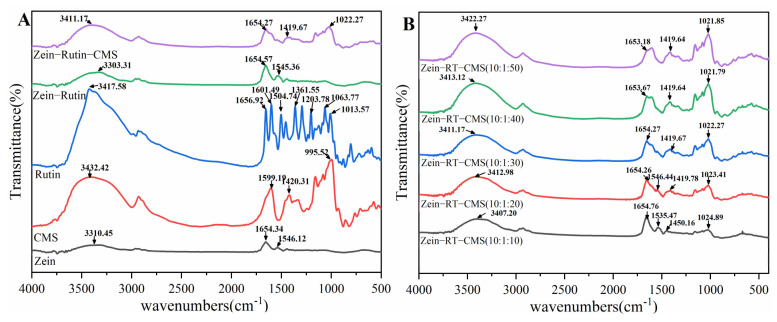
FTIR: (**A**) Comparison of spectrum among zein, CMS, RT, zein–RT, and zein–RT–CMS nanoparticles; (**B**) Spectrum of zein–RT–CMS nanoparticles at different levels of CMS.

**Figure 4 foods-11-02827-f004:**
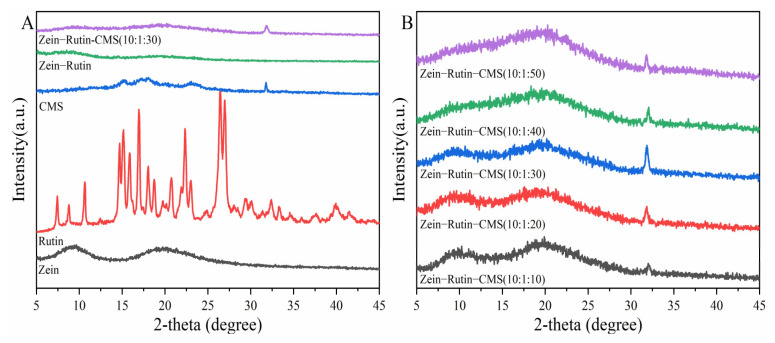
XRD: (**A**) Comparison of XRD among zein, CMS, RT, zein–RT, and zein–RT–CMS nanoparticles; (**B**) XRD of zein–RT–CMS nanoparticles at different levels of CMS.

**Figure 5 foods-11-02827-f005:**
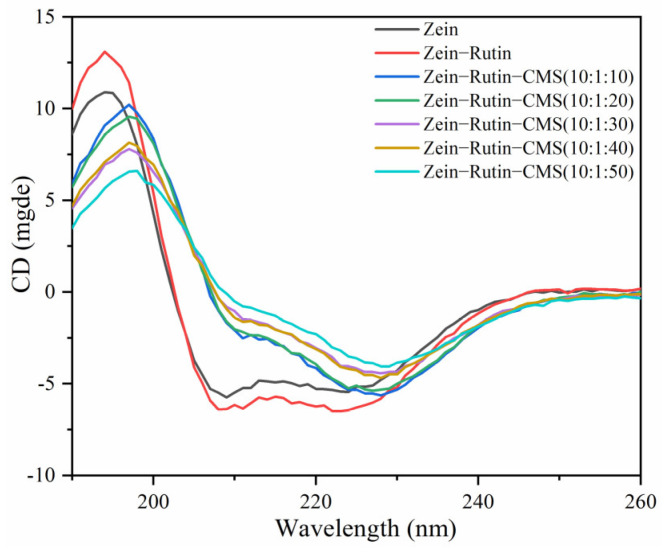
CD spectra of zein and composite particles.

**Figure 6 foods-11-02827-f006:**
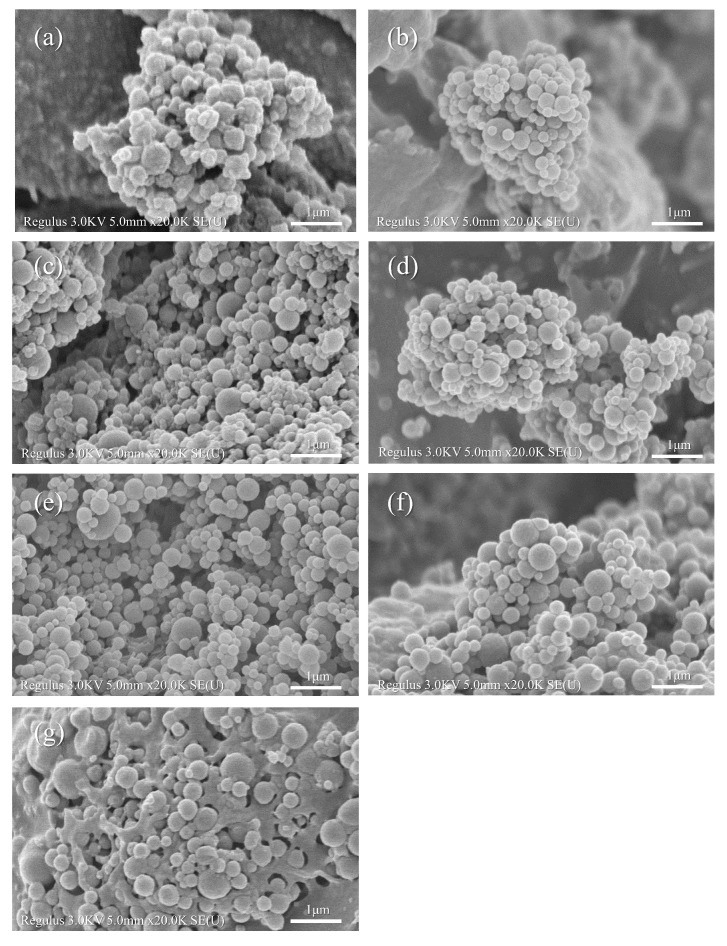
FE–SEM of composite particles: (**a**) zein, (**b**) zein–RT, (**c**) zein–RT–CMS (10:1:10), (**d**) zein–RT–CMS (10:1:20), (**e**) zein–RT–CMS (10:1:30), (**f**) zein–RT–CMS (10:1:40), and (**g**) zein–RT–CMS (10:1:50).

**Figure 7 foods-11-02827-f007:**
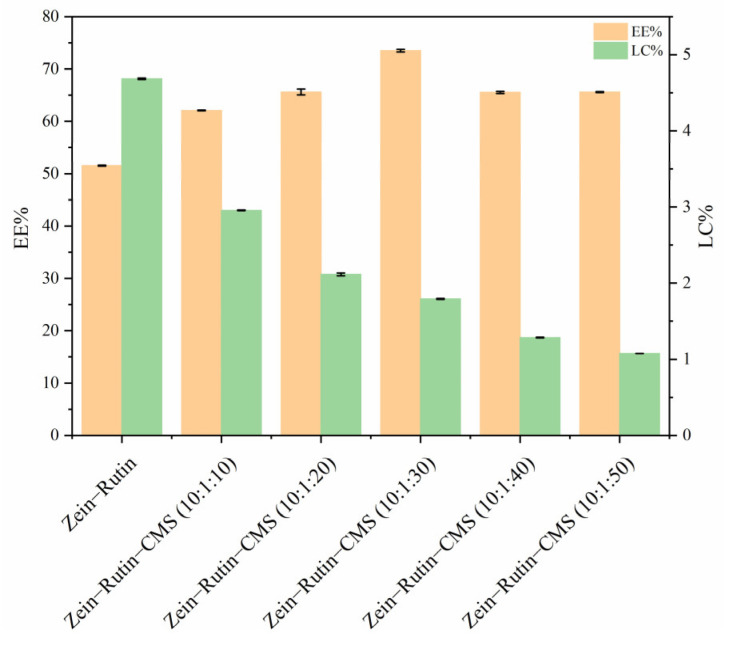
EE and LC of zein–RT nanoparticles and zein–RT–CMS nanoparticles.

**Figure 8 foods-11-02827-f008:**
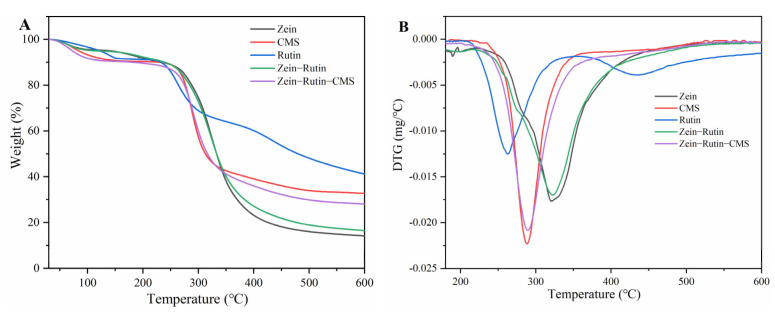
(**A**) TGA spectra of raw materials and composite nanoparticles; (**B**) DTG spectra of raw materials and composite nanoparticles.

**Figure 9 foods-11-02827-f009:**
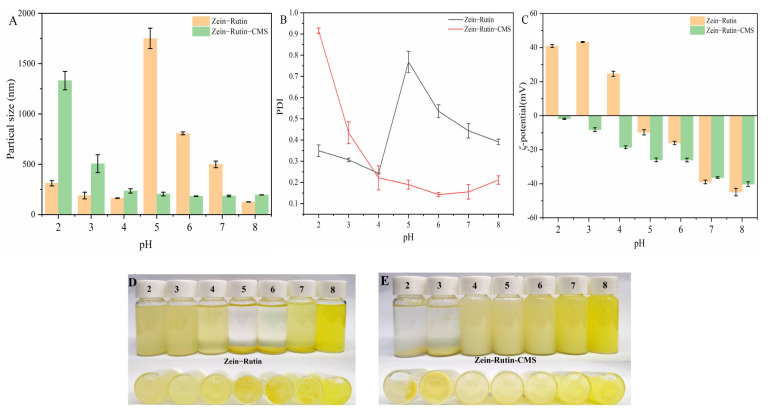
(**A**) Particle size of zein–RT and zein–RT–CMS nanoparticles; (**B**) PDI of zein–RT and zein–RT–CMS nanoparticles; (**C**) ζ–potential of zein–RT and zein–RT–CMS nanoparticles; (**D**) The photograph showed the appearance of zein–RT nanoparticle at different pH; (**E**) The photograph showed the appearance of zein–RT–CMS nanoparticles at different pH.

**Figure 10 foods-11-02827-f010:**
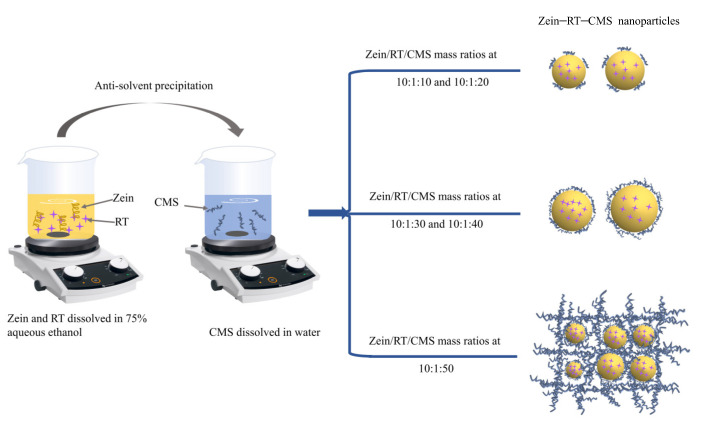
Possible formation mechanism of zein–RT–CMS nanoparticles.

**Table 1 foods-11-02827-t001:** Secondary structure contents of zein in composite nanoparticles.

	Zein	Zein–RT	Zein–RT–CMS (10:1:10)	Zein–RT–CMS (10:1:20)	Zein–RT–CMS (10:1:30)	Zein–RT–CMS (10:1:40)	Zein–RT–CMS (10:1:50)
α–Helix (%)	20.70 ± 0.21 ^e^	21.93 ± 0.12 ^c^	23.70 ± 0.25 ^a^	23.30 ± 0.26 ^b^	22.17 ± 0.10 ^c^	22.07 ± 0.15 ^c^	21.47 ± 0.06 ^d^
β–sheet (%)	28.37 ± 0.31 ^a^	26.67 ± 0.12 ^c^	25.33 ± 0.30 ^e^	25.80 ± 0.29 ^d^	27.20 ± 0.17 ^b^	27.33 ± 0.06 ^b^	28.17 ± 0.12 ^a^
β–Turn (%)	17.93 ± 0.06 ^a^	17.60 ± 0.06 ^e^	16.70 ± 0.12 ^f^	16.80 ± 0.06 ^e^	17.03 ± 0.00 ^d^	17.03 ± 0.06 ^d^	17.20 ± 0.00 ^c^
Random coil (%)	36.67 ± 0.15 ^d^	36.77 ± 0.21 ^d^	40.77 ± 0.30 ^a^	40.67 ± 0.26 ^ab^	40.40 ± 0.10 ^bc^	40.53 ± 0.10 ^abc^	40.33 ± 0.06 ^c^

Note: Different superscript letters among the data in the same row indicate significant differences (*p* < 0.05).

## Data Availability

Not applicable.

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
