# Peer review of "Preparation and Characterization of Rutin–Loaded Zein–Carboxymethyl Starch Nanoparticles"

_foods, 2022, doi:10.3390/foods11182827_

Round 1

Reviewer 1 Report

Though the work appears to be  original and may find applications in food industry, the MS needs extensive revision and improvement in english  language, so that the the sentences   convey the intended  meaning without ambiguity.  The paper is technically sound.

It is concerned  with how they achieved linking of the three substances and the analytical studies on the product. They have used the widely accepted methodology to prepare the zein-CMS-rutin complex and subjected it to FTIR and other studies for characterization. I didn't find anything that needs modification or seek explanation.

I have enclosed my observations on the MS and dispatched them. I am enclosing the MS wherein red marked the language deficiencies.Those in blue are for adding an additional word or revising it. I gave up at the last page as there are mistakes in almost every sentence (with respect to language)

Author Response

Please see the attachment. Thank you again.

Reviewer 2 Report

The authors have studied on “Preparation and characterization of rutin loaded zein/carbox-2 ymethyl starch nanoparticles”. This manuscript describes the use of encapsulation technic on the rutin loaded zein/carbox-2 ymethyl starch nanoparticles to show the efficiency on the Preparation and characterization. They showed that used zein/RT/CMS at 10:1:30 was the best for the encapsulation efficiency. The manuscript is well organized and written and the result could be important for the encapsulation using rutin loaded zein/carbox-2 ymethyl starch nanoparticles. The manuscript needs minor corrections listed below.

Minor points

Line 36-37: Give the vitamin P amount in the fruits, apples and tea.

Line 40-41: what is the starting heat and pH level, please give them.

Line 91: the size of zein should be given.

Line 96: Are the samples prepared under room conditions? If yes, please give room conditions.

Line 123-124: please, give the room temperature and humidity level.

Line 129: what is particle size of the powder?

Line 145: how much samples used?

Line 167: how much samples used?

Line 209-211: this decrease should be discussion with the literature.

Line 342-344: please, explain why 10-1-40 and 10-1-50 (zein-RT-CMS) decrease after the 10-1-30 and discussion with the literature.

Author Response

(The authors gave the same response as above.)

Reviewer 3 Report

In the study by Li et al, rutin (RT) loaded zein-carboxymethyl starch (CMS) nanoparticles were prepared via the antisolvent precipitation method. They studied the effects of different levels of CMS on the composite nanoparticles and evaluated the stability and dispersion of nanoparticles by the addition of CMS. Zein-CMS complex showed a better encapsulation effect on active substances. In addition, they proposed the formation mechanism of zein-RT-CMS nanoparticles. The study was carried out with a good objective and the experiments were well conducted. However, the manuscript is marred with a lot of language errors, and a few are highlighted below. I also recommend that authors carry out a zeta potential study in order to evaluate the charge that develops at the interface between the different formulated composites and the liquid medium.

Page 4, line 120: “and make some modifications” is better as “with some modifications”

Page 5, line 181: add “colon” after “This phenomenon may be caused by the following reasons”. Ie This phenomenon may be caused by the following reasons:

Page 6, line 210 and 228, 233. change “leaded to” to “resulted to”

Others: “ratio at” to “ratio was at”

Page 7, line 266: “hydrogen bonds” should be “O-H bands”

Page 7, line 273: add “colon” after “It is speculated that the possible causes are as follows.”

Increase the size of figures 3a and b.

Delete “scale” in the x axis of figure 4

Page 8, line 281: “XRD was usually used…” should be changed to “XRD is often used ….”

The inscriptions in figure 4 are too tiny

Page 8, line 285: the peak described in the XRD of CMS as wide peak at 32.1° is not wide and should be corrected

Line 286: what is described in the XRD of RT as “spikes” are rather diffraction patterns. The right term should be used

Line 298: “zein-RT nanoparticles [38], it was proved by FTIR analysis” should be written as “zein-RT nanoparticles [38], which agreed with the FTIR analysis”

Give reason for the following observation: From Figure 4B, with the increase of CMS, the characteristic peak intensity of zein-RT-CMS nanoparticles at 32.1° 293 showed a trend of first increasing and then decreasing.

Page 9, line 300: “CD was usually used” should be “CD is often used”

Line 312 to 313: revise the sentence “It can also be explained that some aggregates are caused by β-sheet when the presence of CMS [62].”

Line 314: change “may” to “could”

It is not clear how authors were able to deduce cross-linking network of CMS from the SEM images. This is a mere imagination

Page 11, Line 358: the phrase “The mass loss of the third weightlessness step …” makes no sense

Line 361 to 362. And 363: “loss in the stage of 30 -150 C” should be changed to “loss in the range of 30 -150 C.

Line 376: revise the sentence “can expanded polyphenols application in hot processing food industry”

Increase the size of figures 9a, b and c

The conclusion is too short and should be improved

Author Response

(The authors gave the same response as above.)
